# Targeted Temperature Management after Cardiac Arrest: A Systematic Review and Meta-Analysis with Trial Sequential Analysis

**DOI:** 10.3390/jcm10173943

**Published:** 2021-08-31

**Authors:** Filippo Sanfilippo, Luigi La Via, Bruno Lanzafame, Veronica Dezio, Diana Busalacchi, Antonio Messina, Giuseppe Ristagno, Paolo Pelosi, Marinella Astuto

**Affiliations:** 1Department of Anaesthesia and Intensive Care, “Policlinico-Vittorio Emanuele” University Hospital, 95123 Catania, Italy; luigilavia7@gmail.com (L.L.V.); lanza.bb@gmail.com (B.L.); veronica_dezio@hotmail.it (V.D.); astmar@tiscali.it (M.A.); 2School of Anaesthesia and Intensive Care, University Hospital “G. Rodolico”, University of Catania, 95123 Catania, Italy; diana.busalacchi@gmail.com; 3Department of Biomedical Sciences, Humanitas University, Via Rita Levi Montalcini 4, 20090 Milan, Italy; mess81rc@gmail.com; 4IRCCS Humanitas Research Hospital, 20089 Milan, Italy; 5Department of Anesthesiology, Intensive Care and Emergency, Fondazione IRCCS Ca’ Granda Ospedale Maggiore Policlinico, 20122 Milan, Italy; gristag@gmail.com; 6Anesthesia and Intensive Care, San Martino Policlinico Hospital, IRCCS for Oncology and Neurosciences, 16132 Genoa, Italy; ppelosi@hotmail.com; 7Department of Surgical Sciences and Integrated Diagnostics, University of Genoa, 16132 Genoa, Italy

**Keywords:** cardiac arrest, hospital discharge, neurological outcome, cerebral performance category, mortality

## Abstract

Target temperature management (TTM) in cardiac arrest (CA) survivors is recommended after hospital admission for its possible beneficial effects on survival and neurological outcome. Whether a lower target temperature (i.e., 32–34 °C) improves outcomes is unclear. We conducted a systematic review and meta-analysis on Pubmed and EMBASE to evaluate the effects on mortality and neurologic outcome of TTM at 32–34 °C as compared to controls (patients cared with “actively controlled” or “uncontrolled” normothermia). Results were analyzed via risk ratios (RR) and 95% confidence intervals (CI). Eight randomized controlled trials (RCTs) were included. TTM at 32–34 °C was compared to “actively controlled” normothermia in three RCTs and to “uncontrolled” normothermia in five RCTs. TTM at 32–34 °C does not improve survival as compared to normothermia (RR:1.06 (95%CI 0.94, 1.20), *p* = 0.36; I^2^ = 39%). In the subgroup analyses, TTM at 32–34 °C is associated with better survival when compared to “uncontrolled” normothermia (RR: 1.31 (95%CI 1.07, 1.59), *p* = 0.008) but shows no beneficial effects when compared to “actively controlled” normothermia (RR: 0.97 (95%CI 0.90, 1.04), *p* = 0.41). TTM at 32–34 °C does not improve neurological outcome as compared to normothermia (RR: 1.17 (95%CI 0.97, 1.41), *p* = 0.10; I^2^ = 60%). TTM at 32–34 °C increases the risk of arrhythmias (RR: 1.35 (95%CI 1.16, 1.57), *p* = 0.0001, I^2^ = 0%). TTM at 32–34 °C does not improve survival nor neurological outcome after CA and increases the risk of arrhythmias.

## 1. Introduction

The recently released guidelines for the management of cardiac arrest (CA) patients after return of spontaneous circulation (ROSC) recommend the use of targeted temperature management (TTM) for unresponsive adults after ROSC, regardless the location of CA (in-hospital or out-of-hospital, OH) and the initial detected rhythm (shockable or not) [1].

The TTM should aim at maintaining a constant temperature between 32 °C and 36 °C for at least 24 h, whilst avoiding fever in the first three days after ROSC in patients who remain comatose. After the initial enthusiasm and spread in the use of mild therapeutic hypothermia (MTH, i.e. body temperature lowered to 32–34 °C), a randomized controlled trial (RCT) in 2013 showed no benefits on survival nor on neurological outcome in OH-CA patients treated with MTH over a strategy with TTM at 36 °C [2]. These findings prompted revision of post-resuscitation guidelines towards the broader range of TTM 32–36 °C, and led several hospitals in adopting the higher temperature in this population of patients [3]. However, a subsequent RCT demonstrated that TTM targeting 33°C resulted in higher 90-day survival with favorable neurologic outcome for comatose patients admitted to ICU after (in-hospital and OH-CA with non-shockable rhythm, as compared to TTM at 37 °C [4]. In April 2021, results of the large “TTM 2” RCT on OH-CA patients admitted to ICU were published, showing no differences in survival and in neurological outcome between targeted hypothermia as compared to normothermia with early treatment of fever (≥37.8 °C). In view of the recently published results, we performed a systematic review and meta-analysis to evaluate the effects on mortality and neurologic outcome of TTM with a temperature range of 32–34 °C as compared to controls, defined as CA patients cared with either “actively” controlled or “uncontrolled” normothermia. We also performed a trial sequential analysis to evaluate the statistical power and the potential need for further studies.

## 2. Methods

### 2.1. Search Strategy and Criteria

The meta-analysis was registered on PROSPERO–University of York (registration ID CRD42021233922). We undertook a systematic web-based advanced literature search through the NHS Library Evidence tool on the use of TTM strategies in patients with CA. We followed the approach suggested by the PRISMA statement for reporting systematic reviews and meta-analyses [5] and a PRISMA checklist is provided separately (Appendix A). Our core search was structured by combining the findings from two groups of terms. The first group included the followings: “cardiac arrest” OR “heart attack” OR “cardiopulmonary resuscitation” OR “return of spontaneous circulation”; the second group contained the terms “temperature management” OR “therapeutic hypothermia”. An initial computerized search of PubMed was conducted from inception until 6 January 2021 to identify relevant articles. A final search was re-performed after the publication of the TTM 2 RCT (18 June 2021). We also searched on EMBASE limiting this exploration to the findings published from 2017 in order to retrieve the newest conference abstracts not yet available on Pubmed. Two further searches were performed manually and independently by three authors (L.L.V., F.S., M.A.), exploring also the list of references of the findings of the systematic search.

Inclusion criteria were pre-specified according to the PICOS approach (Table 1). Among studies conducted on OH-CA patients, we included only those where the TTM was implemented and maintained after hospital arrival. Studies focusing only on pre-hospital cooling and no TTM after hospital admission were excluded. We also excluded experimental animal studies, book chapters, reviews, editorials, and letters to editor. Case series were not included in the secondary analysis unless reporting at least 10 patients per group. Language restrictions were applied: we read the full manuscript only for articles published in English. For prospective and retrospective studies published in other languages, we read the abstract and, if necessary, contacted the authors for further information. Study selection for determining the eligibility for inclusion in the systematic review and data extraction were performed independently by four reviewers (L.L.V., B.L., V.D. and D.B.). Discordances were resolved involving one senior author (F.S.). Data were inserted in a password protected database on Excel.

### 2.2. Groups and Endpoints

As primary outcome, we compared the efficacy of TTM strategies on survival and on favourable neurological outcome in RCTs including patients experiencing both in- and/or OH-CA.

We divided our analysis according to the strategy of TTM adopted in controls separating two subgroups: (1) TTM with “actively controlled” normothermia, i.e., avoiding fever, or (2) TTM with “uncontrolled” normothermia (that may hesitate in hyperthermia/fever). As a secondary outcome, we analyzed the incidence of serious adverse events, including only events reported by at least 3 RCTs.

### 2.3. Quality Assessment and GRADE of Evidence

Methodological quality of included RCTs was performed using the Cochrane Collaboration Tool (Centre for Evidence-Based Medicine Odense and Cochrane Denmark, Odense, Denmark) which incorporated the following domains: selection, performance, detection, attrition, performance and other potential sources of bias [6]. Grade of evidence was performed according to the recommendations of the Grading of Recommendations Assessment, Development and Evaluation Working Group by two authors (L.L.V., F.S.) using the GRADEpro software (GRADEpro GDT, Evidence Prime Inc., Hamilton, ON, Canada), available at https://gdt.gradepro.org/ (accessed on 20 June 2021).

### 2.4. Statistical Analysis

The Inverse Variance method was used to analyze dichotomous outcomes of survival at hospital admission and at hospital discharge and survival with good neurological outcome. Results are reported as risk ratio (RR) with 95% confidence intervals (CI) and two tailed *p* values. *p* values were considered significant if <0.05. The presence of statistical heterogeneity was assessed using the X^2^ (Cochran Q) test. Heterogeneity was likely if Q > df (degrees of freedom) suggested and confirmed if *p* ≤ 0.10. Quantification of heterogeneity was performed and values of I^2^ ranging 0–24.9%, 25–49.9%, 50–74.9% and >75% were considered as none, low, moderate and high heterogeneity, respectively. If heterogeneity was quantified as low or above, a random-model was also used for sensitive analyses [7]. Presence of publication bias was investigated by visual inspection of funnel plots for the primary outcomes. We conducted trial sequential analyses (TSAs) in order to evaluate the robustness of our findings, calculating the information size (the power of the meta-analysis) for the survival and favourable neurological outcome at longest follow-up. We used the freely available TSA Software (0.9.5.10 Beta version; Copenhagen Trial Unit’s TSA Software^®^; Copenhagen, Denmark). The information size was computed assuming an alpha risk of 5%, a beta risk of 20%. The estimated effects were computed averaging results of the classical meta-analysis method. Further details on TSA and its interpretation are available elsewhere [8,9].

## 3. Results

Our systematic search identified 3631 findings via NHS Library Evidence search, 2572 of them on Pubmed and 1059 on EMBASE. No other findings were retrieved manually. As shown in the PRISMA flow diagram (Figure 1), after the evaluation of all abstracts, 77 full-text articles were assessed against PICOS criteria. Sixty-nine of them were excluded because of study design: four were prospective but not randomized studies, and 65 included retrospective or historical data. The remaining 8 RCTs were included in our meta-analysis, all reporting data on neurological outcome and survival. The characteristics of the included studies are reported in Table 2. Five RCTs compared “mild therapeutic hypothermia” (target temperature 32–34 °C) to “uncontrolled normothermia” [10,11,12,13,14], but one of them randomized the study population in three groups: hemofiltration with “uncontrolled normothermia”, hemofiltration with “mild therapeutic hypothermia” and no intervention. We included in our meta-analysis only the data from the first two groups of this study [14]. Finally, three RCTs performed an active control of normothermia in the control group [2,4,15].

### 3.1. Survival

The analysis on survival included eight studies, three of them in the subgroup with “active control” of normothermia and the remaining five with “uncontrolled” normothermia (one of them included an 8-h treatment with hemofiltration in both groups). Four studies reported survival at 6 months [10,13,14,15], one at the end of trial (mean period of follow-up was 256 days) [2], one study at 90 days [4], Bernard et al. [11] at hospital discharge, Hachimi-idrissi et al. [12] made the last follow-up 14 days after the randomization.

In the overall analysis, treatment with TTM at 32–34 °C was not associated with improved survival as compared to normothermia: RR 1.06 (95%CI 0.94, 1.20), *p* = 0.36; mild heterogeneity (I^2^ = 39%; Figure 2). The subgroup analyses according to the approach to normothermia in the control group showed significant differences (*p* = 0.005; high heterogeneity, I^2^ = 87%). In particular, TTM at 32–34 °C showed no benefits on survival when compared to “actively controlled” normothermia (RR: 0.97 (95%CI 0.90, 1.04), *p* = 0.41). Conversely, TTM at 32–34 °C was associated with higher survival when compared to “uncontrolled” normothermia (RR: 1.31 (95%CI 1.07, 1.59), *p* = 0.008). 

The exclusion of the study by Laurent et al. [14] did not affect the overall and the subgroup results.

### 3.2. Neurological Outcome

Data on neurological outcome were provided by the same eight studies. Five studies reported neurological outcome at 180 days [2,10,13,14,15], Lascarrou et al. at 90 days [4], Hachimi-idrissi et al. [12] at 14 days and Bernard et al. [11] at hospital discharge.

TTM at 32–34 °C was not associated with improved neurological outcome as compared to normothermia: RR 1.17 (95%CI 0.97, 1.41), *p* = 0.10; mild heterogeneity (I^2^ = 60%; Figure 3). The subgroup analyses according to whether normothermia in the control group was actively pursued or not showed no significant differences (*p* = 0.09; moderate heterogeneity, I^2^ = 65%). TTM at 32–34 °C had no benefits on neurological outcome when compared to “actively controlled” normothermia (RR: 1.02 (95%CI 0.88, 1.17), *p* = 0.79). However, TTM at 32–34°C showed a trend towards better neurological outcomes when compared to “uncontrolled” normothermia (RR: 1.42 (95%CI 1.00, 2.03), *p* = 0.05).

The analysis performed excluding the study by Laurent et al. [14] in which patients received also hemofiltration, modified the overall results. The overall analysis showed a non-significant trend favouring TTM at 32–34 °C (RR 1.20 (95%CI 0.99, 1.46), *p* = 0.06), while the subgroup results showed an improved neurological outcome with TTM at 32–34 °C when compared to “uncontrolled” normothermia (RR 1.50 (95%CI 1.19, 1.89), *p* = 0.0007).

### 3.3. Adverse Events

Adverse events were reported variably by the included studies. We analysed only the incidence of adverse events reported by at least three RCTs, namely bleeding (*n* = 3), pneumonia (*n* = 3), and arrhythmias (*n* = 3). There were no differences in the incidence of bleeding (RR 1.10 (95%CI 0.83, 1.44)) and pneumonia (RR 1.11 (95%CI 0.96, 1.29)) according to the TTM strategy, whilst the incidence of arrhythmias was significantly higher in the patients receiving TTM at 32–34 °C (RR 1.35 (95%CI 1.16, 1.57), *p* = 0.0001, I^2^ = 0%, Figure 4).

### 3.4. Grade of Evidence

The grade of evidence assessed using the GRADEpro software showed that both survival and neurological outcome had moderate certainty. Serious concerns were found only for the inconsistency domain, since there were significant differences in the approach to treatment in the control groups between studies (Appendix A).

### 3.5. Trial Sequential Analysis

The two TSAs performed (one for each primary outcome) included all the RCTs (Appendix A). The TSA on survival according to all RCTs showed that the Z-curve crossed the futility boundaries; therefore, the absence of difference in survival seems robust and no more studies are needed. Conversely, the TSA performed on neurological outcome showed that the Z-curve did not cross the futility boundaries nor the adjusted significance thresholds, meaning that current evidence is not robust enough and thus new research is needed.

### 3.6. Assessing Risk of Bias and Publication Bias

The assessment of risk of bias for RCTs, showed that only the three recent RCTs actively controlled normothermia in the control group [2,4,15] had low risk of bias, while the other studies were at high risk of bias (Appendix A). Therefore, the results excluding studies at high risk of bias would be identical to the subgroup analyses already presented. Visual inspection of funnel plots for both primary outcomes showed absence of publication bias (Appendix A).

## 4. Discussion

This systematic review and meta-analysis showed no beneficial effects of TTM at 32–34 °C in comparison to normothermia, on both survival and neurological outcome. These results were observed when TTM at 32–34 °C was compared to “actively controlled” normothermia with avoidance of fever. Conversely, when compared to “uncontrolled” normothermia, TTM at 32–34 °C improved survival and neurological outcome. The present meta-analysis has several novelties compared to earlier ones. First of all, it includes data only from RCTs in order to assure higher level of evidence and avoiding potential bias related to observational and/or retrospective study designs. Secondly, this review is one of the first including data from the recently released TTM2 trial, together with the Fernando’s one [21], which however mainly investigated effects of deep, mild, and moderate hypothermia vs. normothermia. Most importantly, our meta-analysis focused on the impact of controlled vs. uncontrolled temperature management on CA outcome. Indeed, one of the most important point highlighted by our meta-analysis is that “un-controlled” normothermia has detrimental effects on both survival and neurological outcome. Therefore, our meta-analysis confirms that CA survivors should not be treated with “uncontrolled” normothermia, and clinicians should set a target for an active control of temperature, whether this is hypothermia (TTM at 32–34 °C) or normothermia with fever avoidance. Basically, “uncontrolled” normothermia is not an acceptable strategy nowadays. Regarding the best target of temperature, another meta-analysis focused on this different aspect and included only patients randomized to a TTM strategy. In this meta-analysis the authors showed that mild, moderate, or deep hypothermia do not improve survival or neurological outcome after out-of-hospital CA as compared to normothermia [21]. In our meta-analysis we included only RCTs and in the subgroup analyses there was a clear separation according to the two TTM strategies used as control groups (“uncontrolled” vs. “actively controlled” normothermia). It is worth to note the lag period of almost a decade between RCTs included in these two subgroups. The five RCTs included in the subgroup of “uncontrolled” normothermia were published between 2001 and 2005 [10,11,12,13,14], but after the enthusiasm on therapeutic hypothermia and its growing worldwide application, the TTM-1 trial conducted in 2013 [2] reported the equivalence between TTM at 33 °C versus controlled normothermia at 36 °C. This study led to a change in the recommendations for post-resuscitation TTM management, with a new range-target between 33 °C and 36 °C [1,3]. Since an improvement in bystander-initiated CPR was achieved, the post-CA syndrome has been largely investigated [16], and a standardized post-resuscitation care has been introduced. The proactive post-resuscitation care now recommends hemodynamic and respiratory targets, timely coronary angiography, early TTM and fever avoidance up to 72 h post-ROSC, with introduction of protocols for assessing prognosis and withdrawal of life-sustaining treatment. It is possible that a more comprehensive treatment of CA patients has contributed to ameliorate better survival and neurological outcomes, whilst conversely reducing the positive impact of hypothermia as compared to “actively controlled” normothermia. The recent TTM-2 trial [15] now paves the way towards possible equivalence between hypothermia and normothermia and avoidance of fever, as confirmed by our most recent meta-analysis. Another aspect which may con contribute to the different results between subgroups is related to the methods used to achieve hypothermia. In the earlier RCTs on mild therapeutic hypothermia, temperature reduction was achieved with methods, i.e., ice packs application over the body and head or cool air [10,11,13], which did not allow precise control of TTM with time. Conversely, in the TTM trials [2,15] new feedback-controlled cooling systems have been used, with a more precise temperature control, both during induction and maintenance of TTM as well as during rewarming. These cooling systems have been also used to assure fever prevention over the subsequent 72 h. It is not surprising that greater temperature fluctuations were reported on the RCTs with “uncontrolled” normothermia, in which several patients were often febrile. Fever occurrence is common during the first 72 h after ROSC, and it has been associated with worse outcomes in observational studies [17,18,22]. Whether fever contributes to poor neurological outcome or it is just a marker of severe brain injury remains unknown. However, the TTM-2 trial [15] suggests that avoidance of fever may be a sufficient intervention in the post-resuscitation care, with no need for deeper cooling in the majority of CA patients. Finally, considerations regarding a possible population bias in patients enrolled in the TTM-2 trial should be mentioned. This RCT reported 75% of shockable rhythms, 90% of witnessed CAs, and 80% of bystander-initiated CPR. A similar population was also observed in the TTM-1 trial. In this context, it should be considered that outcome is far better after an initial shockable rhythm [23], and that bystander CPR is associated with a three-fold increase in survival [19]. In the HYPERION trial conducted in a population of non-shockable CAs (with also in-hospital CAs), TTM at 33 °C allowed a better long-term outcome compared to TTM at 37 °C [4]. Unfortunately, the most recent epidemiological data on CA in Europe depict conditions far away from those seen in the TTM-2 trial: 20% of shockable rhythms, 58% of bystander CPR, and 8% survival [23]. Thus, it cannot be excluded that the population in the TTM-2 trial presented a selection bias that might have at least partially halted the beneficial impact of TTM targeting hypothermia as compared to active control of normothermia and avoidance of fever. The negative cardiovascular effects of TTM at 32–34 °C with significantly higher incidence of arrhythmia should be also considered when targeting temperature in unconscious CA patients admitted to ICU. Previous studies reported an association between lower heart rate and good outcome in patients receiving TTM after OH-CA. In a cohort of 111 patients with OH-CA, bradycardia during therapeutic hypothermia was associated with good neurologic outcome at hospital discharge [20]. Similarly, another study on 234 consecutive comatose CA survivors showed that sinus bradycardia during hypothermia had 17% 180-day mortality as compared to doubled incidence in those not presenting sinus bradycardia (*p* < 0.001), as well as lower odds of unfavorable neurological outcome (*p* < 0.01) [24]. Interestingly, one post-hoc analysis of a randomized study showed that bradycardia less than 50 beats/min was independently associated with lower mortality and lower odds of unfavorable neurologic outcome, and that this independent associations were found also in patients maintained at temperature of 36 °C [25]. On the contrary, a higher post-resuscitation heart rate has been associated with in-hospital mortality during the initial 48 h after ROSC [26]. In conclusion, the shift to normothermia is justified according to recent results but abandoning TTM completely is not an option [27].

### Limitations

Our study has several limitations. First, the approach in temperature management in the control group largely varied among trials from “uncontrolled” normothermia with high incidence of fever to an “active maintenance” of normothermia with avoidance of fever. Second, the possible selection bias in the population enrolled in the two recent TTM-1 and TTM-2 studies has been described in terms of very high rates of shockable rhythms as well as witnessed CA and bystander initiated resuscitation [2,15]. Third, our data refers mostly to OH-CA patients with only one RCT [4] including a mixed CA population with around one quarter experiencing in-hospital CA. Fourth, the effects of TTM may be influenced by the initial rhythm of presentation (shockable vs. not) and duration of down-time; these aspects are a matter of debate [28] and probably deserve patient-level meta-analysis. Fifth, the analysis of grade of evidence points towards a moderate certainty of our findings, but the TSA on neurological outcome shows that the findings are not robust and more research is needed; conversely, the equivalence of the different TTM strategies (hypothermia vs. normothermia) in terms of survival seems rather robust with the Z-curve sitting within the futility boundaries.

## 5. Conclusions

In CA survivors admitted to hospital, the implementation of TTM with a target temperature of 32–34 °C does not improve survival nor neurological outcome while it increases the risk of arrhythmias. However, approaching temperature management with “uncontrolled” normothermia is associated with worse outcomes and this should not be considered an option nowadays.

## Figures and Tables

**Figure 1 jcm-10-03943-f001:**
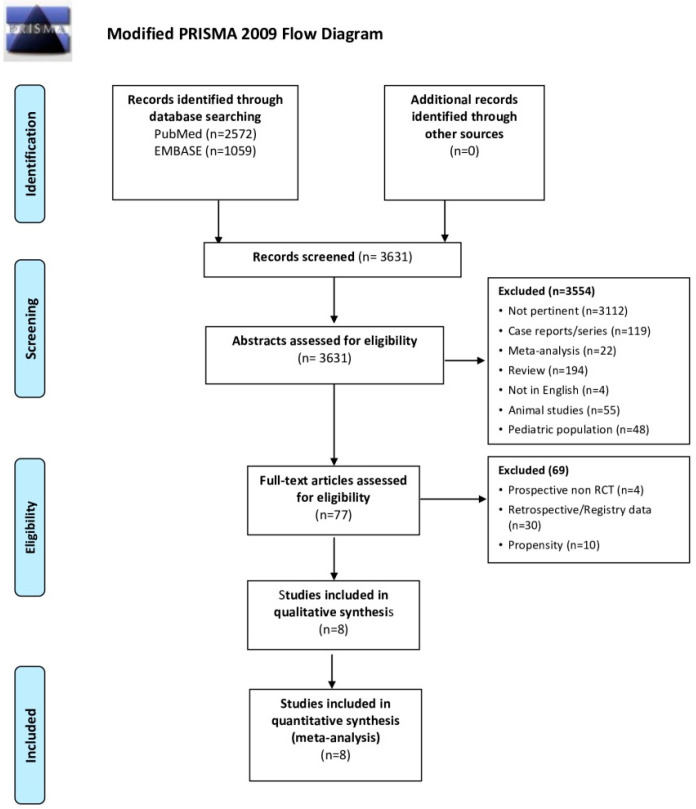
PRISMA flowchart.

**Figure 2 jcm-10-03943-f002:**
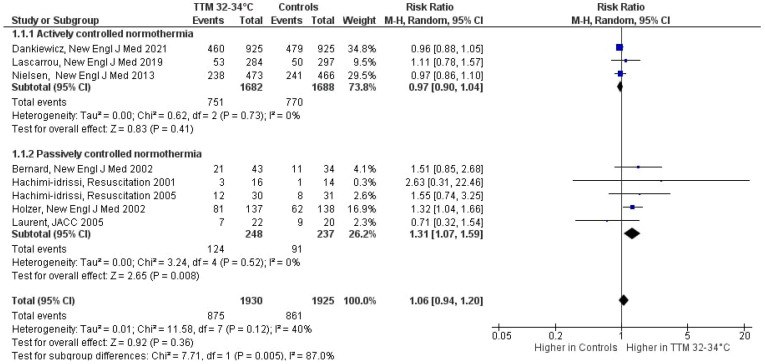
Forest plot of survival in patients resuscitated after cardiac arrest. Com-parison is made between patients according to the strategy of target temperature man-agement (TTM). CI: confidence interval; M-H: Mantel-Haenszel.

**Figure 3 jcm-10-03943-f003:**
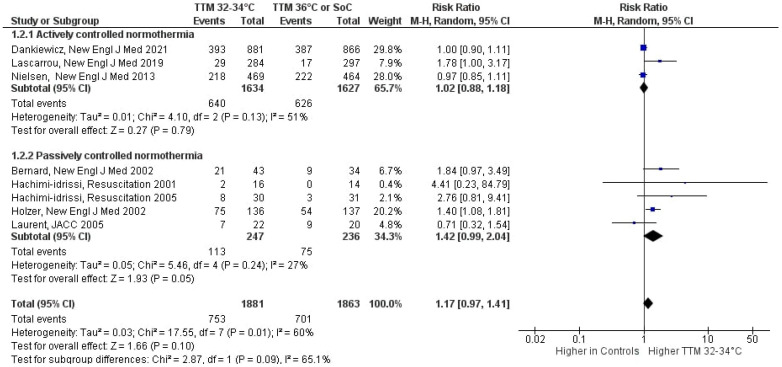
Forest plot of neurological outcome in patients resuscitated after cardiac arrest. Comparison is made between patients according to the strategy of target temperature management (TTM). CI: confidence interval, M-H: Mantel-Haenszel.

**Figure 4 jcm-10-03943-f004:**
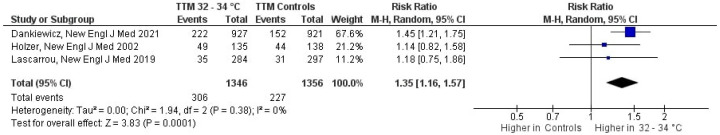
Forest plot of arrhythmias in patients resuscitated after cardiac arrest. Comparison is made between patients according to the strategy of target temperature management (TTM). CI: confidence interval, M-H: Mantel-Haenszel.

**Table 1 jcm-10-03943-t001:** “PICOS” approach for selecting clinical studies in the systematic search and meta-analysis.

PICOS CRITERIA	
Population	Patients experiencing CA both in and out-of-hospital, independently from the initially detected rhythm (shockable or not), with TTM performed after hospital arrival
Intervention	TTM with temperature range set at 32–34 °C
Comparison	TTM with either actively controlled or uncontrolled normothermia
Outcome(s)	Survival and neurological outcome at longest follow-up (primary); adverse effects (secondary)
Study design	Randomized controlled trial only

CA: cardiac arrest; TTM: target temperature management.

**Table 2 jcm-10-03943-t002:** Summary of the included studies of the meta-analysis.

First Author Year	Location of Arrest	First Rhythm Detected	Treatment in the Intervention Group Treatment in the Control Group	Longest Follow Up GNO Assessment	Ref.
Dankiewicz 2021 *N* = 1861	OHCA	Shockable 74% Non-shockable 26%	TTM (surface/ iv, 33 °C, 28 h) + active RW (12 h) Normothermia (≤37.5 °C + surface/iv if ≥37.8 °C)	6-months mRS	[12]
Lascarrou 2019 *N* = 548	Mixed (73% OHCA)	Non-shockable 100%	TTM (any method, 33 °C, 24 h) + active RW (8–16 h, 36 °C, 24 h) TTM (any method, 37 °C, 48 h)	90-days CPC	[15]
Nielsen 2013 *N* = 939	OHCA	Shockable 80% Non-shockable 20%	TTM (any method, 33 °C, 28 h) + active RW (8 h) TTM (any method, 36 °C, 28 h) + active RW (2 h)	6-months—End trial CPC—mRS	[11]
Laurent 2005 * *N* = 42	OHCA	Shockable 74% Non-shockable 26%	TTM (HF + ice-packs, 32 °C, 24 h) + passive RW Normothermia + HF 8 h (37 °C)	6-months CPC	[16]
Hachimi-idrissi 2005 *N* = 61	OHCA	Non-shockable 54%	TTM (Helmet, 33 °C, brief *) + passive RW Normothermia (37 °C)	6-months CPC	[17]
		Shockable 46%	TTM (mattress, 33 °C, 24 h) + passive RW Normothermia (37 °C)		
Holzer 2002 *N* = 136	OHCA	Shockable 96% Other 4%	TTM (mattress, 32–34 °C, 24 h) + passive RW Normothermia (no target)	6-months CPC	[18]
Bernard 2002 *N* = 77	OHCA	Shockable 100%	TTM (ice-packs, 33 °C, 12 h) + active RW (6 h) Normothermia (37 °C)	Hospital discharge Home/short term rehab	[19]
Hachimi-idrissi 2001 *N* = 30	OHCA	Non-shockable 100%	TTM (Helmet, 34 °C, brief *) + passive RW Normothermia + treatment of fever (38 °C)	2-weeks CPC	[20]

HF: hemofiltration; OHCA: out-of-hospital cardiac arrest; RW: rewarming; TTM: target temperature management; mRS: modified rankin scale; CPC: cerebral performance category. * The control group not treated with HF was not considered (*n* = 19).

## Data Availability

The data presented in this study are available on request from the corresponding author.

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
