# Peer review of "Targeted Temperature Management after Cardiac Arrest: A Systematic Review and Meta-Analysis with Trial Sequential Analysis"

_jcm, 2021, doi:10.3390/jcm10173943_

Round 1

Reviewer 1 Report

Thank you very much for having been involved in the revision of this meanalysis. The topic is important, the paper is well written and the statistical methods are solid. My major concern is about the need of a further metanalis on this issue. Many similar paper are present in literature so I would like the Authors to stress in the discussion section the real importance of their paper and what they do add to modern knowledge about TTM. While reading the paper the reader have to understand the importance and the novelty of their results.

Reviewer 2 Report

The metaanalysis study investigated the effects of TTM on survival and neurological outcomes in cardiac arrest patients. The patient groups were separated into TTM, controlled normothermia and uncontrolled normothermia. The analysis was sound and comprehensive.

Comments:

  1. The results of the study is not surprising. The TTM1 and TTM2 trials have higher patient numbers, which would have major impacts on the results of meta-analysis.
  2. As the authors described, there was about 10 more years time lag between "controlled" and " uncontrolled" normothermia studies. How about the outcomes of treatment groups (TTM groups) between studies with these time lag?
  3. The patient population was cardiac arrest. However, the outcomes were heterogeneous for patients with different resuscitation scenarios , such as initial rhythm, no flow time, low flow time, pre-cooling conscious level. The mixing of these characteristics would diminish the effects of TTM treatment, which could be beneficial or useless for some specific groups. At least, the concept should be discussed in the Discussion or Limitation. 

Round 2

Reviewer 1 Report

The Authors adequately answered to my comments.

Author Response

We thank the reviewer for the comment.